# Evaluation of Non-Alcoholic Beverages and the Risk Related to Consumer Health among the Romanian Population

**DOI:** 10.3390/nu15173841

**Published:** 2023-09-02

**Authors:** Magdalena Mititelu, Carmen-Nicoleta Oancea, Sorinel Marius Neacșu, Gabriel Olteanu, Alexandru-Tiberiu Cîrțu, Lucian Hîncu, Theodora Claudia Gheonea, Tiberius Iustinian Stanciu, Ion Rogoveanu, Fallah Hashemi, Gabriela Stanciu, Corina-Bianca Ioniță-Mîndrican, Caunii Angelica, Nicoleta Măru, Sergiu Lupu, Carmen Elena Lupu

**Affiliations:** 1Department of Clinical Laboratory and Food Safety, Faculty of Pharmacy, “Carol Davila”, University of Medicine and Pharmacy, 020956 Bucharest, Romania; magdalena.mititelu@umfcd.ro (M.M.); gabriel.olteanu@mst.umfcd.ro (G.O.); alexandru-tiberiu.cirtu@mst.umfcd.ro (A.-T.C.); 2Department of Biochemistry, Faculty of Medicine, University of Medicine and Pharmacy from Craiova, 200345 Craiova, Romania; carmen.oancea@umfcv.ro; 3Department of Pharmaceutical Technology and Bio-Pharmacy, Faculty of Pharmacy, Carol Davila University of Medicine and Pharmacy, 020945 Bucharest, Romania; neacsusorinelmarius@gmail.com; 4Department of Drug Industry and Pharmaceutical Biotechnologies Department, Faculty of Pharmacy, University of Medicine and Pharmacy Carol Davila, 020956 Bucharest, Romania; lucian.hincu@umfcd.ro; 5Center for IBD Patients, Faculty of Medicine, University of Medicine and Pharmacy from Craiova, 200345 Craiova, Romania; ionirogoveanu@gmail.com; 6Press Office, Ovidius University of Constanța, 900527 Constanța, Romania; 7Department of Environmental Health Engineering, School of Health, Shiraz University of Medical Sciences, Shiraz 71348-14336, Iran; info.foo@gmail.com; 8Department of Chemistry and Chemical Engineering, Ovidius University of Constanta, 900527 Constanta, Romania; gstanciu@univ-ovidius.ro; 9Department of Toxicology, Faculty of Pharmacy, “Carol Davila” University of Medicine and Pharmacy, 020945 Bucharest, Romania; corina-bianca.ionita-mindrican@drd.umfcd.ro; 10Department of Drug Analysis and Chemistry of Environmental Factors, Hygiene, Nutrition, Faculty of Pharmacy, “Victor Babes” University of Medicine and Pharmacy of Timisoara, 300041 Timisoara, Romania; caunii.angelica@umft.ro; 11Department of Anatomy, Faculty of Dental Medicine, “Carol Davila” University of Medicine and Pharmacy, 020945 Bucharest, Romania; nicoleta.maru@umfcd.ro; 12Department of Navigation and Naval Transport, Faculty of Navigation and Naval Management, “Mircea cel Batran” Naval Academy, 900218 Constanta, Romania; sergiu.lupu@anmb.ro; 13Department of Mathematics and Informatics, Faculty of Pharmacy, “Ovidius” University of Constanta, 900001 Constanta, Romania; clupu@univ-ovidius.ro

**Keywords:** sweetened drinks, energy drinks, metabolic diseases, sedentariness, public health, food addiction

## Abstract

The range of non-alcoholic drinks is very varied both from a compositional point of view and from a caloric and nutritional point of view. The excessive consumption of sweetened non-alcoholic beverages represents an important risk factor for health, especially when it is accompanied by an unbalanced diet and a disordered lifestyle. In order to evaluate the consumption of non-alcoholic beverages correlated with the evaluation of the main lifestyle factors that can affect the state of health among Romanians, a cross-sectional observational study was carried out based on a questionnaire. The results of the study indicate that among the most consumed non-alcoholic drinks are coffee and sweetened carbonated and non-carbonated drinks, which are indicated as being responsible for the development of consumption addictions: 44% for coffee, 16.5% for sweetened or tonic carbonated drinks and 12% for sweetened non-carbonated drinks. Considering that the consumption of coffee is usually associated with sweeteners, there is a risk of excessive caffeine and caloric intake in a context where a lack of exercise predominates (59.98%) among respondents declaring that they do sports rarely or not at all, which can lead, in the long term, to the appearance of imbalances either of a psycho-emotional nature or of a metabolic nature. A significant link was found between sports activity and the environment in which they work (χ^2^ = 51.33, *p* = 0.05). Respondents with a daily activity that involves movement (working outdoors, working on a construction site) are also those who usually do sports, while 60.67% of the respondents who work a lot in front of the computer declared that they do sports very rarely or not at all. Reducing the excessive consumption of sweetened drinks can be achieved through an appropriate consumption of water and fruits and by intensifying physical activity as a way of counterbalancing the excess caloric intake.

## 1. Introduction

Non-alcoholic beverages represent an important category of liquids, some with beneficial properties for the consumer, depending on the composition. The most representative of these are mineral waters, tea, coffee, vegetable and fruit juices, lemonades, sweetened carbonated drinks, and energy drinks. Natural non-alcoholic drinks contribute to the mineralization and hydration of the body, and natural juices contain vitamins, antioxidants, carbohydrates, and enzymes (the unpasteurized ones). The energy value of non-alcoholic beverages is different and depends mainly on the carbohydrate content [1,2,3,4]. The nutritional value of natural drinks is determined by the raw material used to obtain them, but also by their quality. It is best to use fruits and vegetables that come from areas with low pollution, because heavy metals, microplastics, fertilizers and pesticides in the soil and air can contaminate the raw material and endanger the safety of the consumer [5,6,7,8].

Some of the most consumed non-alcoholic beverages are coffee and tea (green or black). It has been scientifically proven that moderate consumption of these drinks brings a series of important benefits to the body (Figure 1) through the contained phytonutrients (many with antioxidant action) alongside substances with a psycho-stimulating effect (caffeine, theobromine, theophylline) that increase alertness and eases intellectual work. The recommended dose of coffee is calculated taking into account the percentage of caffeine in the composition [9,10,11,12].

The FDA (Food and Drug Administration) recommends a maximum caffeine dose allowed per day of 400 mg for a healthy adult, but each person has a certain degree of sensitivity to coffee [13]. Excessive caffeine consumption (Figure 2) can cause behavioral disorders (like tremors, anxiety, irritability, nervousness, etc.), exhaustion or chronic fatigue (by reducing the sleep duration or quality, causing insomnia), can increase the risk of stroke by increasing blood pressure, or can disrupt the absorption of iron or calcium [14,15]. Likewise, excessive consumption of green or black tea can produce the same adverse effects.

Sweetened soft drinks and energy drinks are among the most unhealthy non-alcoholic drinks. Sweetened carbonated drinks are hyper-caloric due to the very high content of carbohydrates and, at the same time, they also contain many synthetic additives. Excessive consumption of sweetened carbonated drinks represents a risk for obesity, diabetes, and cardiovascular or gastrointestinal complications (Figure 3). Fructose represents the carbohydrate with the greatest risk for dyslipidemia and gout. The high content of fast-absorbing carbohydrates in these drinks generates, in the long term, an increase in insulin resistance, but also an addiction to consumption. Carbonated diet drinks generally contain synthetic sweeteners that have proven to be quite toxic to the body, some with carcinogenic potential in addition to hepatotoxicity [16,17,18,19,20,21,22].

Energy drinks have become a very popular type of drink around the world, being consumed mostly by teenagers and young adults between the ages of 18 and 34. There are many reasons why young people choose to consume energy drinks. Some teenagers and students consume them to combat sleepiness, to have more energy, or in combination with alcohol at parties. Other young people admit to drinking these drinks for no particular reason or when they feel exhausted [23,24].

The main active ingredient in energy drinks is represented by caffeine, to which other ingredients are frequently added, such as sugar, taurine, glucurono-lactone, glucose, guarana (*Paullinia cupana*), vitamins from the B complex, and artificial sweeteners. Excessive caffeine consumption is known to cause insomnia, increase blood pressure and lead to osteoporosis and cardiovascular disease over time (Figure 4). Moreover, the high carbohydrate content of most energy drinks is another health hazard. A small 250 mL dose of energy drink often contains between 21 and 34 g of sugar, or even higher doses of up to 60 g, which can cause weight gain and tooth decay. One of the reasons why type 2 diabetes is very widespread in the world is the excessive consumption of drinks that combine sugar and caffeine [25,26].

Some researchers indicate that consuming energy drinks increases the risk for alcohol dependence in people who regularly drink alcohol or even in those who have previously undergone rehab and also have associated the consumption of energy drinks, as quite common among young people, with specific behavioral problems, such as drinking alcohol, smoking, or consuming other prohibited substances [27,28,29].

The combination of alcohol with energy drinks can become dangerous in large doses because it impairs judgment about the level of intoxication and leads to what is called “lucid drunkenness” [30,31].

This practice can have extremely dangerous consequences for the individual, such as increasing the risk of drunk driving, alcohol intoxication, dehydration and even death. The FDA has already banned the sale of energy drinks premixed with alcohol in the US, but the law does not prohibit individuals or bartenders from making this mix [32,33].

The excessive consumption of sweetened non-alcoholic drinks correlated with an unhealthy diet and lifestyle represents a risk factor for the population and is especially a promoter of metabolic syndrome with complications [34].

Unfortunately, in Romania, obesity and the risk of diabetes are constantly increasing. The Global Obesity Observatory classified Romania’s national obesity risk as 7 out of 10, indicating a high risk based on the prevalence of obesity [35]. Moreover, these carbonated drinks, through packaging, marketing, and the taste generated by sweeteners, are consumed in excess by children and adolescents, who are not fully aware of the harmful effects they are exposed to. In addition to the consumption of fast food, the consumption of sweets and sweetened carbonated drinks are the basis of childhood obesity which is considered “one of the silent pandemics of the 21st century” [36].

In this sense, at the beginning of 2023, the Romanian Government increased the value-added tax (VAT) in the case of sweetened carbonated drinks (mineral waters and carbonated waters containing sugar or other sweeteners, soft drinks, energy drinks, coffee, tea, cappuccino, soy-based drinks or other plant-based beverages such as syrups) from 9% to 19% [37]. This amendment to the legislation by Ordinance no. 16 of 15 July 2022 comes after all the alarming statistics regarding the risk of consumption of these products on the health of the population (increased amount of sugar and sweeteners, low nutritional value, risk of addiction). In Romania, the situation is alarming, because the consumption of sweetened non-alcoholic beverages per capita increased from 209.8 L in 2018 to 234 L in 2021 [38].

As a result, a study was carried out based on a questionnaire that tracked the consumption of non-alcoholic beverages correlated with lifestyle (physical activity, duration and quality of sleep, hydration of the body, periodic assessment of health status, association with alcohol and tobacco consumption) but also with body weight and problems affecting well-being among the Romanian population.

## 2. Materials and Methods

### 2.1. Study Design

The cross-sectional observational study carried out on the basis of a questionnaire with 49 questions followed the evaluation of the consumption of non-alcoholic beverages correlated with the main lifestyle factors that can affect health and well-being (physical activity, sleep quality, body hydration level, tobacco use, health monitoring) among the Romanian population 18 years old and above. In relation to the consumption of non-alcoholic beverages, the main aim was to evaluate the frequency of consumption and the number of different types of drinks, the use of sweeteners and their type in different drinks such as tea, coffee, lemonades, the highlighting of consumption addiction through the perception of the respondent and the identification of consumption preferences for the main types of non-alcoholic beverages, as well as the eventual association of these with non-alcoholic beverages. The questionnaire was structured into three main parts: the first part aimed at obtaining socio-demographic and anthropometric information (age, gender, residence, level of education, occupational status, weight, and height); the second part aimed at collecting data related to the frequency of consumption of different categories of non-alcoholic beverages, the amount consumed, and the types of associated sweeteners; and in the part of the third was the collection of information related to lifestyle (physical activity, tobacco consumption, alcohol consumption, duration and quality of sleep, periodicity of health assessment, type of professional activity), a series of aspects that alter the quality of life (the state of the immune system, the presence of conditions such as fatigue, nervousness, depression, anxiety, migraines, agitation, palpitations, compulsive eating or loss of appetite), as well as the identification of consumption addictions generated by some non-alcoholic beverages. The dissemination of the questionnaire was conducted with the help of the Google Forms web platform in the online environment through social networks, but also with the help of WhatsApp and institutional emails (within higher education units or companies that agreed to participate in the study) between March and June 2023. Participation was on a voluntary basis and aimed at people 18 years old and above, without any discrimination related to sex, religion, or political beliefs. The data collected by the questionnaire tracked information related to sex, age, occupation, residence, education, height, weight, eating habits, lifestyle, and diagnosed conditions. The design and dissemination of the questionnaire were carried out in accordance with the requirements for ensuring the confidentiality of the information (without the collection of email addresses or personal data that would identify the respondents), and the completion of the questionnaire was carried out with the informed consent of the study participants.

### 2.2. Questionnaire Validation

The questionnaire was validated with the help of a group of 6 experts who also tested it in the pilot phase on a group of 250 respondents to establish the final form of the questionnaire and increase its accuracy and clarity. The Cronbach’s α coefficient was determined to assess the internal consistency of the questionnaire, and for the final version, the value of 0.87 was established (corresponding to good internal consistency) [39,40,41,42].

### 2.3. Statistical Analysis

Descriptive statistics was used to present the background characteristics of participants. Categorical variables are presented with absolute frequencies (*n*) and relative frequencies (%).

To identify potential associations between preferred non-alcoholic drinks and anthropometric data (Age and BMI), simple correspondence analysis was applied. Differences in categorical variables were analyzed by using a chi-square test.

After chi-square tests, we used column proportion z-tests to determine the relative order of the column categorical variables in terms of the proportions of the row categorical variables to determine which rows and columns were responsible for the relationship. Results of the statistical processing of the consumption frequency of soft drinks are based on two-sided tests. Finally, we analyzed the association between the outcome variable ”Quantity of non-alcoholic beverages” with independent variables such as gender, age groups, and BMI groups by applying logistic regression for a multinomial model, and the results were expressed by odds ratio (OR), interval confidence (95% IC) and the associated *p* values. The odds ratio was changed and explained in percentages. Statistical analysis was performed using XLSTAT (version 2020, Addinsoft, New York, NY, USA) for Correspondence and Statistical Package Software for Social Science, version 23 (SPSS Inc., Chicago, IL, USA), and *p*-values of less than 0.05 were considered statistically significant [43,44,45].

## 3. Results

### 3.1. Socio-Demographic and Anthropometric Data

The study carried out based on the dissemination of the questionnaire registered a total number of 1754 valid answers that came from 81.4% female respondents and 18.6% male respondents. With the anthropometric data, we calculated the BMIs of respondents by using the Quetelet equation (body mass (kg)/height (m^2^)) and interpreted them according to the criteria of the World Health Organization [46,47,48]. For a good accuracy of the clinical results, BMI must be correlated with the percentage of fat, because there are situations in performance athletes when a higher BMI does not represent an excess of adipose tissue but a higher percentage of muscle mass. In the case of a predominantly sedentary behavior, excess weight is accompanied by excess adipose tissue. After processing the anthropometric data, it was observed that most of the respondents are of normal weight (54.4%), in particular women (56.9% of normal-weight respondents). Most of the obese (13.8% of total respondents) and overweight (24.8%) respondents are male, and most of the underweight respondents (7% of total respondents) are female (8.2% of female respondents). Most of the respondents (54.3%) come from among the young population aged up to 35 years. The socio-demographic and anthropometric characteristics of the respondents are presented in Table 1.

### 3.2. The Types of Drinks Frequently Consumed Correlated with the Demographic and Anthropometric Variables

The analysis of the frequency of consumption of different types of non-alcoholic drinks is very important because many of these drinks have a consistent caloric load, even the natural ones. Table 2 presents data on the caloric intake and the content of sweeteners in the composition of some sweet non-alcoholic drinks. As can be seen, fresh juices also bring a fairly significant caloric load and are close to sweetened soft drinks; they have the advantage of containing a series of valuable nutrients (vitamins, mineral salts, antioxidants, and enzymes in natural form), but related to consumption, it is necessary to monitor the caloric intake of these drinks as well.

In our study, the data collected for the analysis of the frequency of consumption of the main categories of non-alcoholic drinks (green tea, black tea, herbal tea, coffee, hot chocolate, cocoa, sweetened or non-carbonated tonic drinks, sweetened or carbonated tonic drinks, syrups, lemonades, natural juices, energy drinks, non-carbonated mineral water, carbonated mineral water) were quantified as follows: 1 for frequent consumption and 0 for total lack of consumption.

Both chi-square analysis and correspondence analysis (CA) were conducted for the anthropometric variables BMI and age.

The bi-plot indicated that for the BMI group, 94.94% of the variability observed could be attributed to the two main components (F1: 80.16%, F2: 14.77%). CA showed a significant difference (χ^2^ = 58.12 and *p* = 0.0058) between BMI and the consumption of the 14 types of drinks in the survey (Figure 5), but not between the frequency of consumption of the analyzed drinks and respondents with normal BMI and underweight. There is an increased tendency to consume sweetened carbonated drinks, syrups, and carbonated mineral water among obese and overweight people.

The bi-plot for the age group indicated that 96.89% of the variability observed could be attributed to the two principal components (F1: 86.43%, F2: 11.38%). CA showed a significant difference (χ^2^ = 54.57 and *p* < 0.0001) between age groups and the 14 types of drinks in the survey (Figure 6). Regarding the consumption of coffee, there was no significant difference in consumption between the different age groups, while the age groups showed significant differences in consumption preference compared to the other types of drinks. The younger generation, aged between 18 and 25, prefers energy drinks, juices, and green tea more, while the 26–35 age group prefers black tea, lemonade and carbonated drinks. The 36–50 age group prefers mineral water and the over 50 age group prefers herbal tea.

### 3.3. Frequency and Quantity of Soft Drinks Consumption

The first outcome variable for this analysis was the frequency of non-alcoholic and alcoholic beverage consumption. The upper case letters (A, B, C, D) indicate significance levels for BMI and (A, B) for gender (α = 0.05). Tests are adjusted for all pairwise comparisons within each suitable innermost row using the Bonferroni correction (Table 3). A comparison of column proportions indicated that the percentage of people who drink coffee over 36 years old is significantly higher than that of younger people (up to 35 years old).

The percentage of people over the age of 50 who consume Coca-Cola or Pepsi very rarely or not at all is significantly higher (78%) than that of younger people. Regarding the consumption of green tea, there were no significant differences between the age groups and the frequency of consumption. The majority of respondents over the age of 50 (93.1%) declared that they consume black tea very rarely or not at all—a significantly higher percentage than those under the age of 36.

The percentage of young people between the ages of 18 and 25 who consume sweetened carbonated drinks more than two times a week is significantly higher than that of the other age groups.

The comparisons of column proportions indicated that the percentage of women who consume coffee daily is significantly higher than that of men; on the other hand, male respondents tend to consume more of the other analyzed beverage categories.

In addition, the frequency of consumption of alcoholic beverages was analyzed and it was found that the respondents in general declared that they used to consume alcohol either a few times a week or a few times a month (less than daily), and the same trend was noted in all age groups and with a higher consumption tendency among male respondents.

An association between frequency of consumption and BMI was found only for coffee (χ^2^ = 31.16 and *p* = 0.002), as we can see in Figure 7, and sweetened carbonated drinks (χ^2^ = 34.61 and *p* = 0.003), as presented in Figure 8. The z-test indicated that the percentage of people with a BMI > 25 (74% of overweight respondents and 71.3% of obese respondents) who consume coffee daily is significantly higher than those with normal weight. The way coffee is consumed also comes into play here—especially the carbohydrate content, which can significantly influence caloric intake.

Regarding sweetened carbonated drinks, the data indicate a significantly higher percentage of obese respondents (13.6%) consume more than one portion daily (Figure 8). The percentage distribution of the BMI group regarding the frequency of consumption of sweetened non-alcoholic drinks also indicates that the highest percentage of people who rarely consume these soft drinks is found among normal-weight respondents (50.3%). The influence of these types of drinks on body weight is manifested not only by the frequency of consumption, but also by the caloric content and the type of sweetener in the composition of the drinks, which can be significantly different; it is also closely related to the level of daily physical activity of the consumers.

The second outcome variable for this analysis was the quantity of non-alcoholic beverages. To apply multinomial logistic regression, the quantity was coded as follows:-For coffee, black tea and green tea consumption “Rarely or not at all”, per day (reference group), “1 cup” per day, “2–3 cups” per day and “More than 3 cups” per day;-For Coca-Cola or Pepsi “Rarely or not at all”, per day (reference group), “up to 500 mL” per day, “up to 1 L” per day and “Over 1 L” per day;-For energy drinks “Rarely or not at all”, per day (reference group), “1 dose” per day, “2 doses” per day, “over 2 doses” per day.

The multinomial logistic regression models are displayed only for significant (*p* <  0.05) independent variables, anthropometric variables (sex, age and BMI) and lifestyle variables (sport, sleeping time and smoking).

No association was found between the amount of green tea or black tea consumed and independent variables.

The amount of coffee consumed by respondents was found to be associated with the independent variables age (*p* < 0.0001), gender (*p* = 0.005), BMI (*p* = 0.002) smoking status (*p* < 0.0001) and also with sleeping time (*p* < 0.0001) (Table 4).

According to the processed results, women are most likely to consume daily a cup of coffee (OR = 1.776) or two to three cups per day (OR = 1.547) compared to men (Table 4).

The frequency of daily coffee consumption increases with age; thus, the very young respondents (age group 18–25) show the lowest consumption tendency, both regarding a single cup a day (OR = 0.220), as well as consumption of two to three cups a day (OR = 0.150). Respondents over 50 years of age show the highest trend related to the frequency and quantity of coffee consumed. Regarding the BMI groups, there is an increase in the consumption trend related to the frequency and quantity of coffee consumed proportional to weight gain. Underweight respondents have the lowest consumption tendency, while overweight respondents have the highest tendency to consume both a single cup of coffee per day (OR = 1.959) and two to three cups per day (OR = 3.056).

Respondents who smoke, even occasionally, are more likely to consume large amounts of coffee than non-smokers.

Excessive coffee consumption can affect the quality of sleep; respondents who have frequent insomnia are more likely to drink more than three cups of coffee per day (OR = 6.075) compared to people who sleep 7–9 h a night. The most pronounced tendency to consume two to three cups of coffee a day or even more is the people who sleep fewer hours a night or have repeated episodes of insomnia.

In total, 42.42% of respondents have the habit of drinking sweetened coffee. The normal-weight respondents consume sweetened coffee most frequently (53.09% of them), but the type and amount of sweetener used, as well as the level of physical activity, obviously matter (Figure 9a). The amount of sweetened coffee consumed, as well as the total consumption of carbohydrates per day, especially those with fast absorption, also matters. Regarding the frequency of sweetened coffee consumption, the highest trend is noted among very young people (18–25 years old) and the lowest trend for people over 50 years old (15.05% of them) (Figure 9b).

The most used sweeteners are white sugar (25.54%), honey bee (23.77%) and brown sugar (18.93%) (Figure 10). These categories are especially preferred by young people between 18 and 25 years old (Figure 11). With the exception of bee honey, the other types of sweeteners considered healthier alternatives to sugar are consumed in very low proportions among the respondents participating in the study. Chi-square tests showed no statistical significance between types of sweeteners and BMI.

Older respondents, especially those over 50 years old (Figure 11), tend to consume the most variants of sweeteners with lower caloric intake than sugar which are considered much healthier (xylitol, stevia, agave syrup, maple syrup). Most young people (47.32% of those aged between 18 and 25) prefer white sugar as a sweetener.

According to the recorded answers, 41.71% of the respondents used to drink coffee before the meal. Among them, normal-weight respondents most frequently consume coffee before meals (Figure 12a) and those aged between 36 and 50 (Figure 12b).

The quantity of Coca-Cola and Pepsi consumed was found to be associated with the independent variables gender (*p* = 0.008), age (*p* < 0.0001), BMI (*p* = 0.001), smoking status (*p* < 0.001) and sleeping time (*p* < 0.001) (Table 5).

Regarding the amount of Coca-Cola or Pepsi drinks consumed per day, there is a tendency to consume a larger amount among male respondents than female respondents (OR = 0.668), especially in quantities greater than 500 mL.

Unlike coffee, sweetened carbonated drinks are consumed in larger quantities, especially by young people. Thus, the tendency to consume a large amount of Coca-Cola or Pepsi daily decreases with age. Respondents aged up to 25 showed the highest consumption tendency. Regarding the amount of sweetened carbonated drinks consumed, we notice a progressive increase with weight gain (overweight and obese respondents tend to consume much larger quantities compared to underweight ones). We also note increased consumption trends among respondents who smoke, but also among those who have frequent insomnia or sleep less than 7 h a night.

The number of energy drinks consumed was found to be associated with the independent variables age (*p* < 0.0001), gender (*p* = 0.001) and smoking status (*p* = 0.002) (Table 6). In the case of energy drinks, there is also a tendency to consume a larger amount among men compared to women.

Also, the tendency to consume energy drinks decreases with age and increases with the habit of smoking—those who smoke excessively having the highest tendency to consume energy drinks. Thus heavy smokers tend to consume double or triple the amount of energy drinks compared to non-smokers (OR = 3.168 for 1 dose per day).

Regardless of the consumption of soft drinks, the hydration of the body must be properly ensured through water intake, and the consumption of calories provided by food, especially sweets and sweetened drinks, must be realised by physical activity. As a result, the questionnaire followed the evaluation of the appropriate hydration of the body, but also of the physical activity of the respondents. There is a fairly large percentage (Figure 13) of respondents who consume a maximum of 1L of water per day (44.69%).

A significant association was found between the amount of water and the habit of moving (χ^2^ = 160.61, *p* < 0.0001). Approximately 80% of respondents who consume less than 1 L of water per day either do not do physical activity or do it very rarely (Figure 14).

The CA bi-plot for sport frequency and main type of non-alcoholic drinks indicated that 81.58% of the variability observed could be attributed to the two principal components (F1: 64.80%, F2: 16.79%). The horizontal axis of doing sport daily for at least one hour is in contrast with not doing sport, and has a similar profile with doing sport two to three times a week. The bi-plot suggests associations of not doing sport with energy drinks, sweetened or tonic carbonated drinks, and carbonated mineral water. Doing sport daily for under an hour is associated with the consumption of lemonade, herbal tea, and natural juices (Figure 15).

Unfortunately, the consumption of sweetened beverages is not accompanied by a corresponding movement (Figure 16), with most respondents declaring that they do sports rarely or not at all (59.98% of them).

The preferred sports activities are those outdoors (Figure 17); very few of the active respondents choose to do sports in the gym (20.20%).

A significant link was found between sports activity and the environment in which they work (χ^2^ = 51.33, *p* = 0.05). Respondents with a daily activity that involves movement (working outdoors, working on a construction site) are also those who usually do sports, while, unfortunately, those who have a daily activity with little movement have the greatest tendency towards sedentarism. Thus, 60.67% of the respondents who work a lot in front of the computer declared that they do sports very rarely or not at all, and 59.53% of those with an office or with a predominantly static job declared the same (Figure 18).

Regarding the state of well-being (Figure 19), only 27% of the respondents declared that they do not encounter any problems, while 51.8% frequently feel tired, 23.20% are frequently nervous, 20.40% eat excessively emotionally, 19% are frequently agitated and 16.5% are frequently depressed.

The main factors that respondents claim are the most dangerous for their health (Figure 20) are stress (73.6% of respondents feel stressed), sleep quality (50%), lack of exercise (44%), and food quality (42.2).

Among the most indicated drinks that generated addiction among respondents (Figure 21) are coffee (44% of respondents), sweetened non-carbonated drinks (12%), and sweetened carbonated drinks (16.5%).

## 4. Discussion

This study was carried out to analyze the consumption of non-alcoholic beverages and included a total of 1754 respondents, most of whom (54.3%) were between 18 and 35 years old (Table 1). According to the socio-demographic data, the majority of respondents live in the urban environment (80.3% of them) and have higher education (60.9%), are women (81.4%), and commute daily to work (43.7%). Although most respondents are young, 38.6% of them are overweight or obese. The category most likely to consume sugar-sweetened beverages is children and adolescents [65]. Marketing, the availability of non-alcoholic drinks sweetened with sugar in various bars, cafes, shops, and vending machines, and also the pressure from the environment are some of the factors that influence young people to consume products with poor nutritional value. Also, adolescents are prone to adopt a dietary behavior that endangers health (consumption of fast food, sugar-sweetened soft drinks, sedentary lifestyle, sleep deprivation) [66,67].

According to the collected data, the most consumed categories of non-alcoholic drinks are coffee (73.1% of respondents), lemonades (34.7%), natural juices (33.1%), and sweetened carbonated drinks (32.2%). All these drinks come with a series of sweeteners that can increase caloric intake and addiction (Table 2).

Sweetened carbonated drinks are especially consumed with fast food. In recent years, variants of diet drinks have also appeared on the market, such as Diet Coke. One of the thousands of chemicals used in these new foods is monosodium glutamate (MSG or E 621) [68]. MSG is a food additive with the function of enhancing flavor, the use of which has increased considerably, being found in many processed foods that fill the supermarket shelves. Through its action on glutamate receptors and the release of neurotransmitters, it can play an important role in the emergence of pathological processes, the most documented of which is the onset of obesity (by altering the action of leptin, increasing palatability, insulin resistance, release of pro-inflammatory mediators), neurotoxicity (Alzheimer’s disease, Huntington’s disease, Parkinson’s disease, epilepsy, multiple sclerosis, and brain tumors) and reproductive system damage (oxidative stress and protein and DNA changes) [69,70].

In addition to MSG, there are a host of other sweeteners incriminated for their addictive effects and cellular metabolic imbalances, namely high-fructose corn syrup [71], aspartame, acesulfame K (Ace-K) [72], sucralose [73], and cyclamate [74]. There are other sweeteners considered non-toxic such as saccharin [75] and stevia [76].

Coffee is the most indicated by the respondents as being responsible for addiction (Figure 21); in total, 66.4% of them consume either one cup (40% of consumers) or two to three cups (32.5%) daily. Approximately 75% of the respondents consume green or black tea very rarely or not at all. Herbal teas are especially preferred and especially by respondents over 35 years old.

Regarding the consumption of coffee, over 59% of the respondents consume it frequently or sometimes sweetened, and over 65% of them usually associate it with milk. The main sweetener used is sugar, which is especially preferred by young people up to 35 years old (Figure 10 and Figure 11). This usually involves a caloric addition to the coffee both by combining it with sweeteners and by combining it with milk.

The most consumed assortment of coffee is the one with caffeine (83.8% of the respondents) and, as a rule, coffee consumption is associated with Coca-Cola, Pepsi, or even energy drink consumption, especially in young people and smokers (Table 5 and Table 6). The effects of these associations are reflected in the quality of sleep (39% of respondents have insomnia or sleep less than 7 h a night), but especially in the factors indicated by respondents that affect their well-being (Figure 19), including fatigue, agitation or nervousness, and emotional overeating.

The literature is rich in evidence correlating the consumption of sweetened non-alcoholic beverages with an increased risk of mortality, especially from cardiovascular causes [76]. It found that consumption of two or more sugar-sweetened non-alcoholic beverages per day resulted in a 31% increased risk of death from cardiovascular disease (RR 1.31, 95% CI 1.15–1.50) compared to zero or less than one sugar-sweetened drink per month [77]. Yin et al. conducted a meta-analysis in 2021 that studied a number of six cohort studies, finding that for each one-serving-per-day increase in the consumption of sugar-sweetened beverages, the risk of cardiovascular mortality increases by 8% (RR 1.08, 95% CI 1.04–1.13) [77].

Regarding the link between these drinks and the risk of type 2 diabetes, Imamura et al. conducted a meta-analysis of 17 prospective cohort studies and concluded that an increment of one serving per day of sugar-sweetened beverages was associated with an 18% higher risk of developing type 2 diabetes [78]. Type 2 diabetes is a major public health problem because the patient is in permanent danger due to the significant structural and functional changes and imbalances that this pathology exerts on the human body (microvascular complications: diabetic retinopathy that can lead to blindness, neuropathy and nephropathy; macro-vascular complications: coronary artery disease, stroke and peripheral arterial disease—extremely dangerous because it progresses from ecchymoses, wounds that heal poorly or not at all, gangrene to amputation) [79].

The analysis of the physical activity of the respondents unfortunately highlighted a high degree of tendency towards sedentarism (Figure 16), a worrying fact considering that most of the respondents were young. Moreover, the consumption of sweetened drinks leads to an excessive caloric intake if physical activity is reduced, and in the long term, it leads to excess weight and the accumulation of adipose tissue with serious consequences for health. This fact is also reflected in the 18.98% of obese and overweight people among young people up to 35 years old. In the long term, there is a risk of developing metabolic syndrome if physical activity is not intensified and the consumption of carbohydrates with fast assimilation—those with low molecular weight—is not reduced. Unfortunately, the situation is not good when it comes to the optimal hydration of the body with water, with many of the respondents consuming amounts of up to 1 L of water per day, which are insufficient amounts. Water is particularly important for the detoxification of the body, the digestion of food and the optimal absorption of nutrients, but also for the proper functioning of nerve cells [80,81]. Well-being, attention, and power of concentration are maintained by rest, exercise, and adequate hydration [82,83]. The state of well-being and implicitly the quality of life is affected; although most of the respondents are up to 35 years old, they declared that they face a series of problems such as fatigue, nervousness, agitation, and depression, with very few of the participants in the study declaring that they feel well (Figure 19). Moreover, many declared that they were stressed (Figure 20). All these aspects, especially the excessive consumption of carbohydrates combined with reduced physical activity, affect work capacity, integration into the community, and, of course, health [84,85,86]. Regarding the state of health, 19% of the respondents declared that they have a weakened or unbalanced immune system, 21.5% resort to different methods to strengthen immunity, only 37.6% evaluate their state of health once a year, 16.9% of the respondents declared that they hardly ever evaluate their health status, and 22.5% rarely evaluate their health status.

In this context, government policies to limit the excessive consumption of drinks with added sweeteners are welcome and important for the prevention of obesity and type 2 diabetes in particular.

The study has a series of limitations determined by the low presence of male respondents, those from rural areas, and people over 50 years old. It is important, however, to analyze the consumption of sweetened beverages among the young population, because this represents the main active labor force and the basis for ensuring future generations. The results of the study managed to capture some important aspects related to the mistakes that can affect the health of consumers in the future.

## 5. Conclusions

The consumption of non-alcoholic drinks raises health problems when they bring about too high an intake of carbohydrates with fast absorption due to the caloric excess or too high an intake of synthetic sweeteners with a high glycemic index, but also when they provide an excess of caffeine or of other neuro-excitant substances. In the context of a lack of physical activity, excess calories and carbohydrates can lead to long-term metabolic problems (obesity, dyslipidemia, type 2 diabetes), and excess caffeine primarily disrupts the quality of sleep, can alter the psycho-emotional balance, and can aggravate a series of cardiovascular diseases. Also, the addiction generated by a series of non-alcoholic drinks (coffee, sweetened carbonated and non-carbonated drinks) must also be managed, as excessive consumption often takes place at the expense of proper hydration with water, which can lead to an aggravation of imbalances in the body of consumers (dehydration, improper detoxification, blood thickening, etc.). Managing the consumption of sweetened beverages must be carried out from a young age to prevent the development of excessive consumption habits. The results of the study indicate a significant addiction to coffee consumption, but also an increased frequency of states of fatigue and agitation generated by inadequate rest and probably by excess caffeine. In order to counterbalance the negative effects of excessive consumption of carbohydrates and caffeine, proper hydration of the body and stimulation of physical activity are required.

## Figures and Tables

**Figure 1 nutrients-15-03841-f001:**
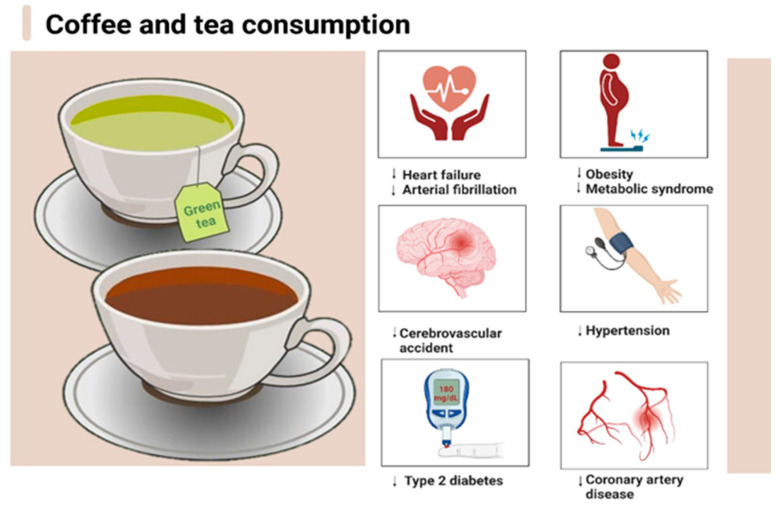
The benefits of moderate tea and coffee consumption. Created with BioRender.com.

**Figure 2 nutrients-15-03841-f002:**
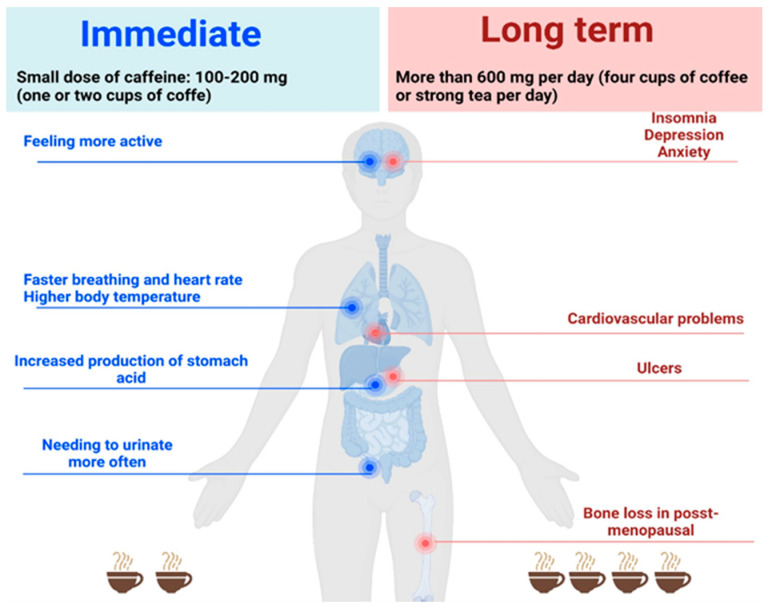
The effects of coffee consumption depending on the amount of caffeine ingested. Created with BioRender.com.

**Figure 3 nutrients-15-03841-f003:**
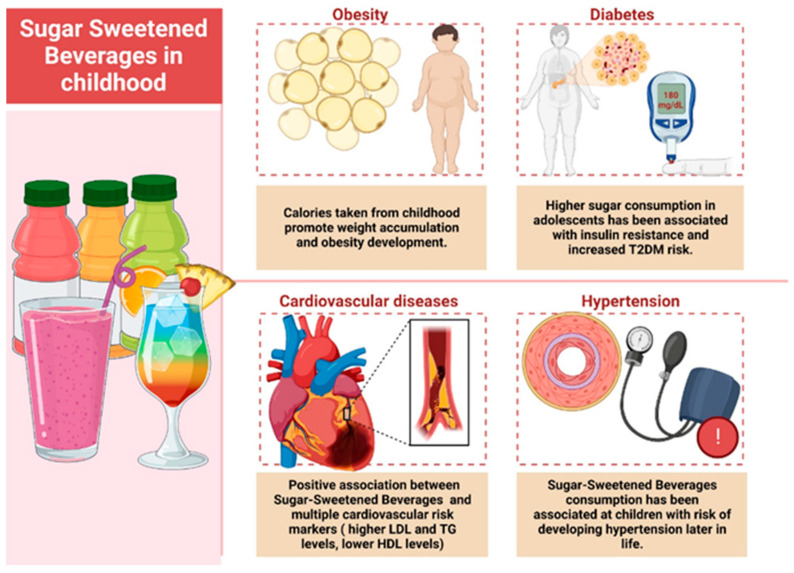
The effects of excessive consumption of sweetened drinks. Created with BioRender.com.

**Figure 4 nutrients-15-03841-f004:**
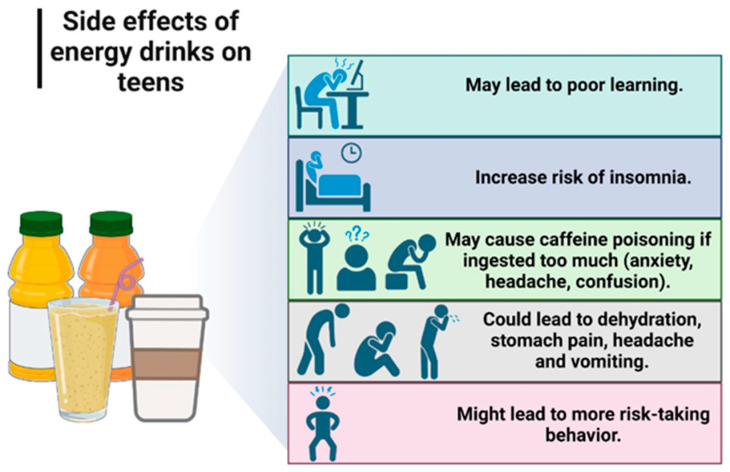
The effects of excessive consumption of energy drinks. Created with BioRender.com.

**Figure 5 nutrients-15-03841-f005:**
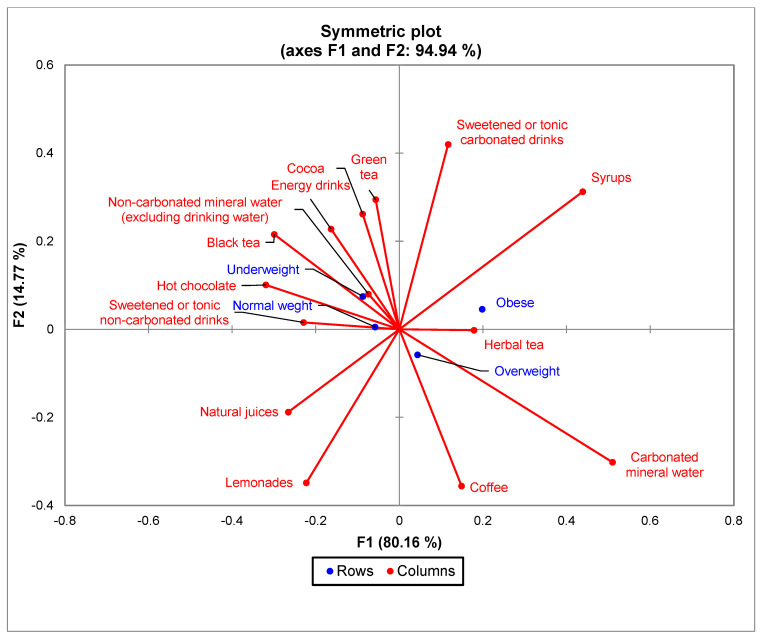
The first two dimensions of correspondence analysis (CA) symmetric plot using BMI groups and all the 14 types of drinks analyzed.

**Figure 6 nutrients-15-03841-f006:**
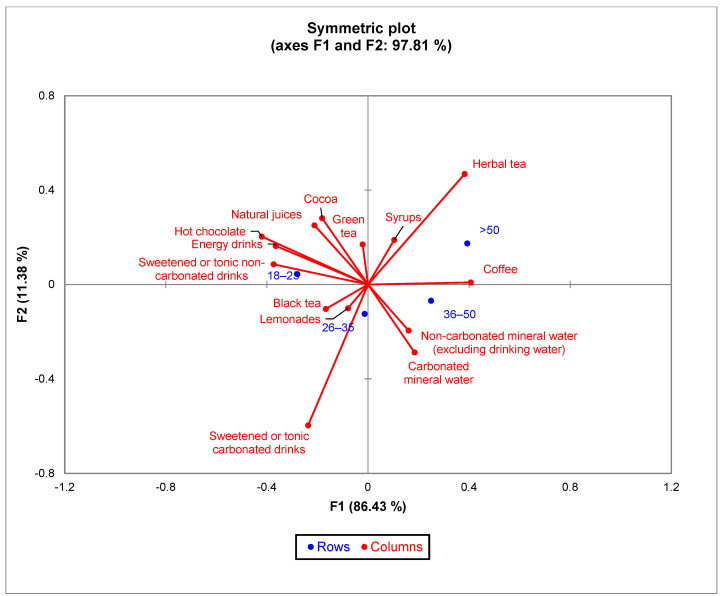
The first two dimensions of correspondence analysis (CA) symmetric plot using age groups and all the 14 types of drinks analyzed.

**Figure 7 nutrients-15-03841-f007:**
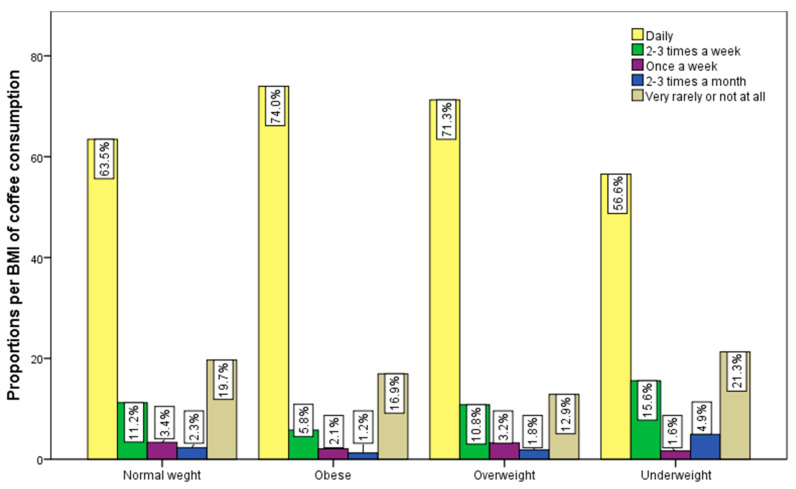
Frequency of coffee consumption and BMI groups.

**Figure 8 nutrients-15-03841-f008:**
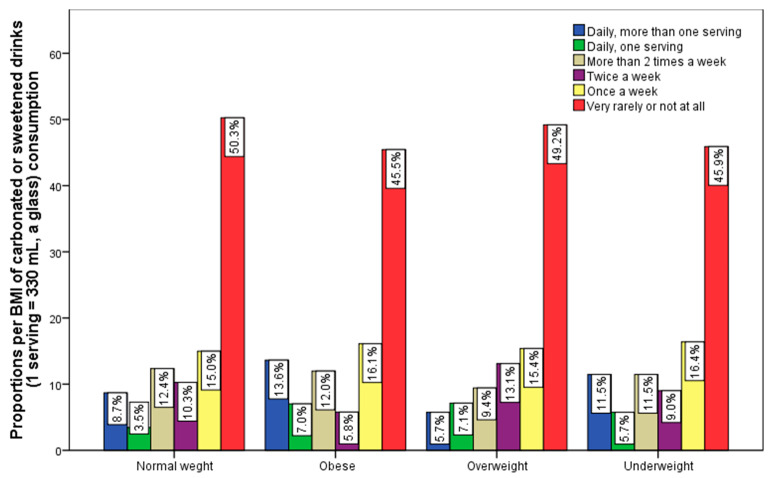
Frequency of carbonated or sweetened drinks consumption and BMI groups.

**Figure 9 nutrients-15-03841-f009:**
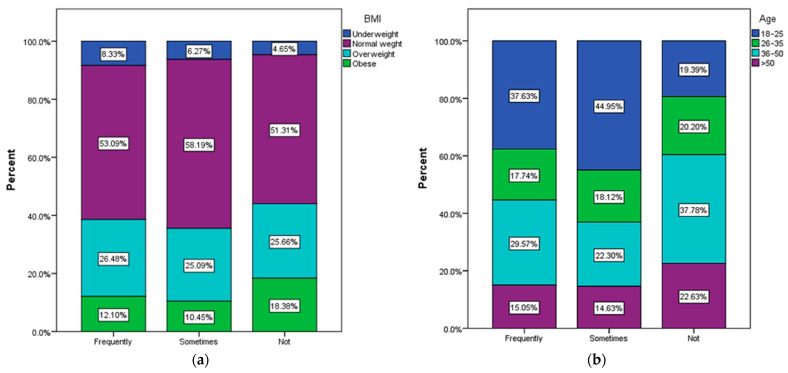
Habits of drinking sweetened coffee by (**a**) BMI and (**b**) Age.

**Figure 10 nutrients-15-03841-f010:**
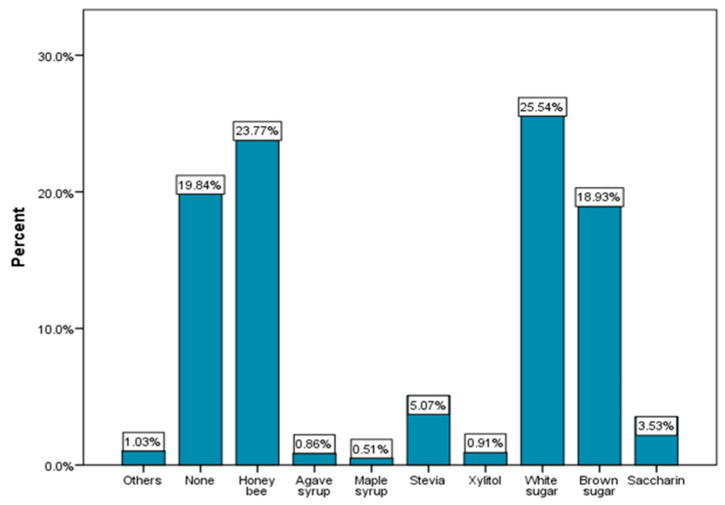
Type of sweetener used the most for non-alcoholic beverages.

**Figure 11 nutrients-15-03841-f011:**
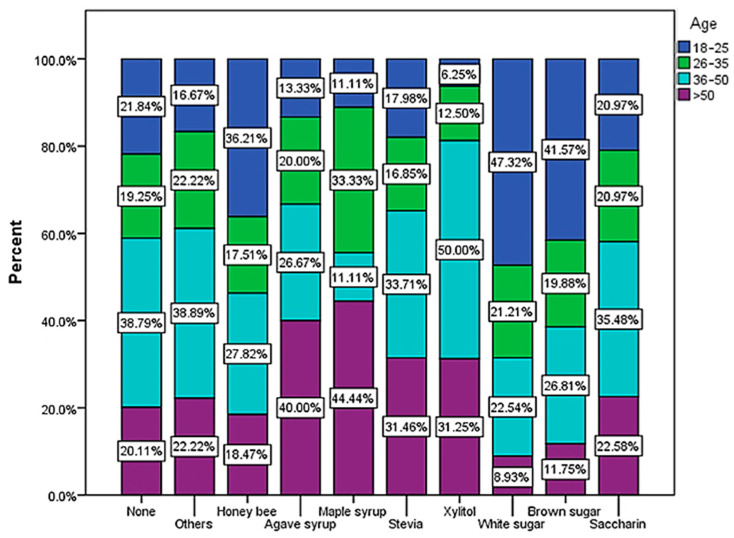
Type of sweetener used the most for non-alcoholic beverages by age (χ^2^ = 49.46 and *p* = 0.005).

**Figure 12 nutrients-15-03841-f012:**
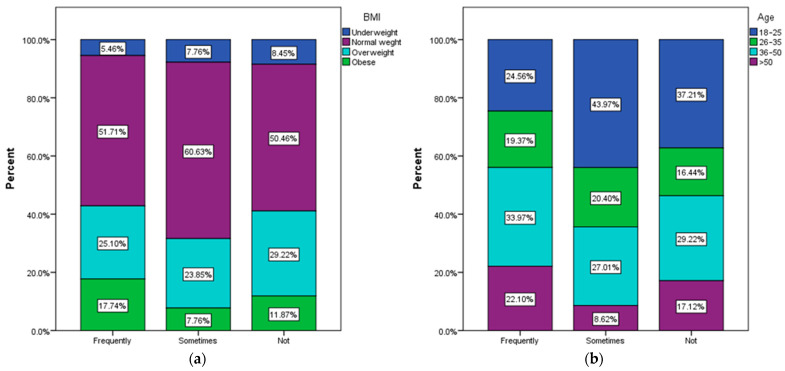
Habits of drinking coffee before meal by (**a**) BMI and (**b**) age.

**Figure 13 nutrients-15-03841-f013:**
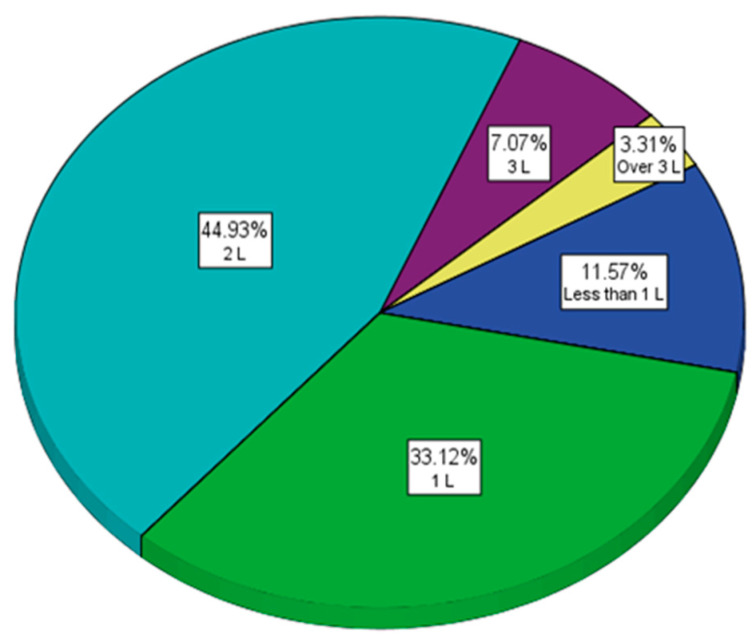
The amount of water consumed daily.

**Figure 14 nutrients-15-03841-f014:**
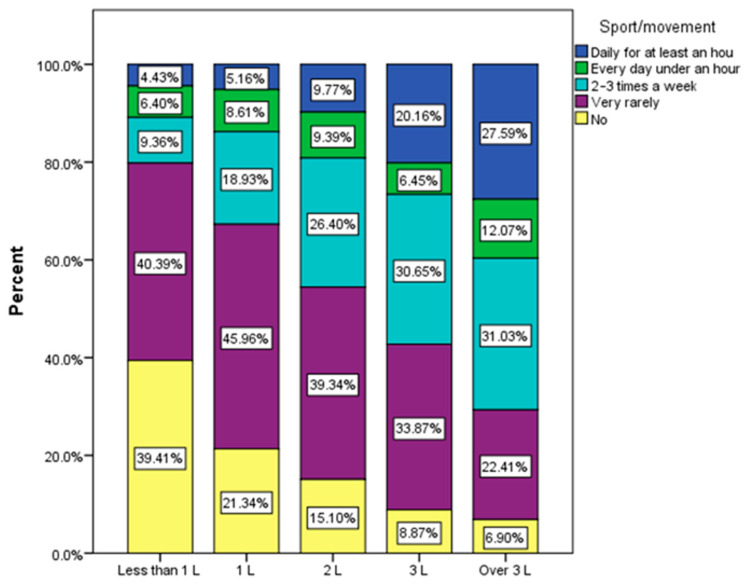
Quantity of drink water per day correlated with sport/movement.

**Figure 15 nutrients-15-03841-f015:**
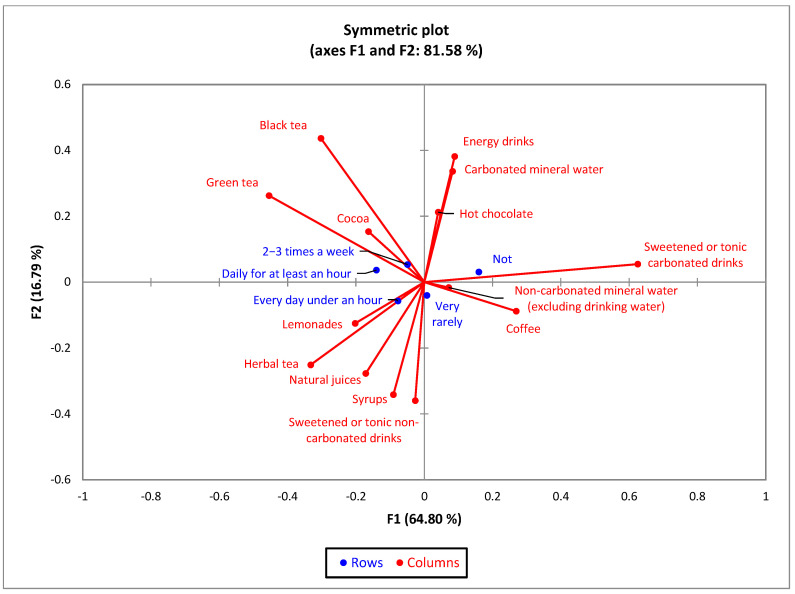
The first two dimensions of correspondence analysis (CA) symmetric plot using frequency of sports activity and all the 14 types of drinks analyzed.

**Figure 16 nutrients-15-03841-f016:**
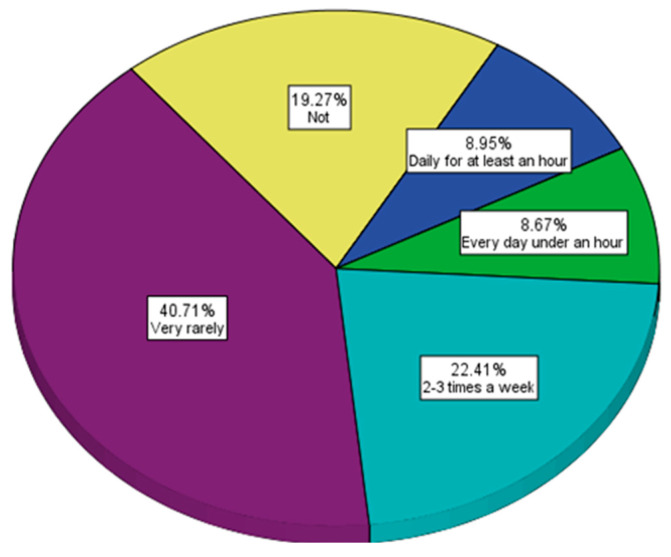
Frequency of sports activities.

**Figure 17 nutrients-15-03841-f017:**
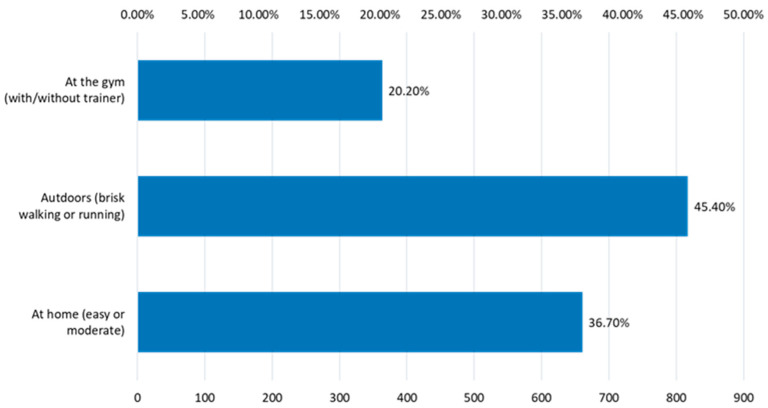
The location of the sports activities.

**Figure 18 nutrients-15-03841-f018:**
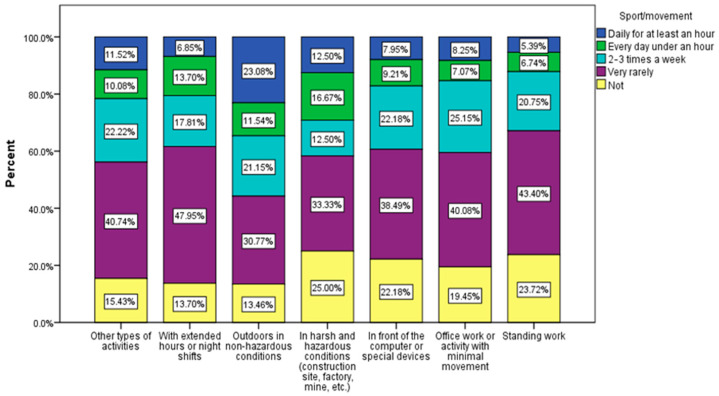
Type of work associated with sport/movement.

**Figure 19 nutrients-15-03841-f019:**
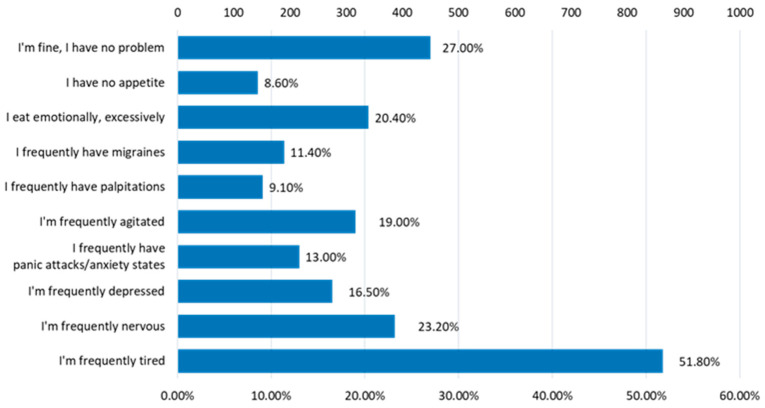
Problems that affect well-being.

**Figure 20 nutrients-15-03841-f020:**
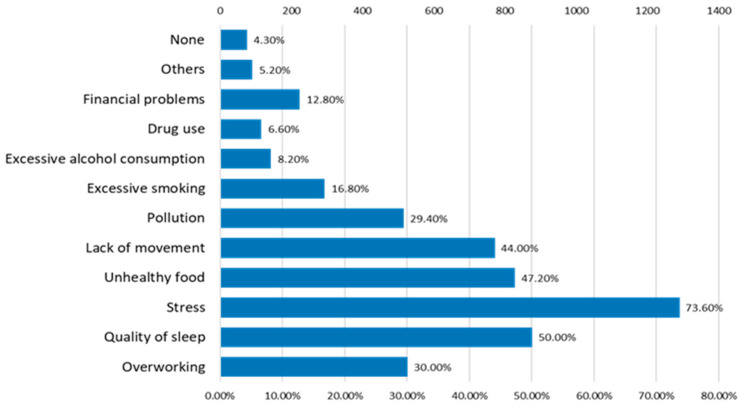
The factors that affect the state of health.

**Figure 21 nutrients-15-03841-f021:**
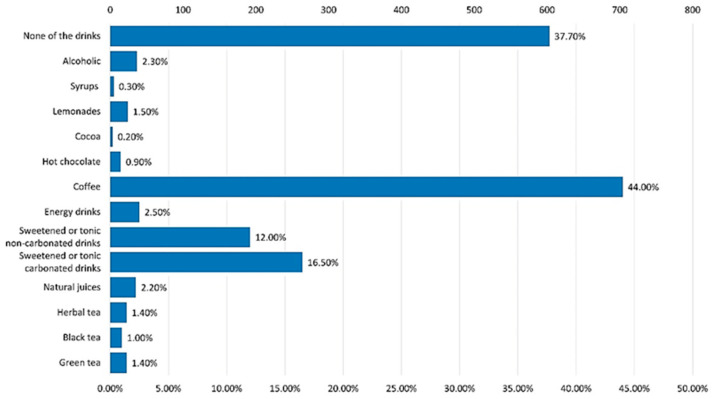
Addiction to non-alcoholic beverages.

**Table 1 nutrients-15-03841-t001:** Socio-demographic and anthropometric characteristics of respondents (*n* = 1754).

	Total Population *n* (%)	Male *n* (%)	Female *n* (%)
	1754 (100)	327 (18.6)	1424 (81.4)
Age (years)		*p* < 0.0001	
18–25	613 (34.9)	158 (48.3)	455 (31.9)
26–35	341 (19.4)	50 (15.3)	291 (20.4)
36–50	513 (29.2)	73 (22.3)	440 (30.8)
>50	287 (16.4)	46 (14.1)	241 (16.9)
Residence areas		*p* = 0.673	
Urban areas	1409 (80.3)	268 (82.0)	1141 (80.0)
Rural areas	166 (19.7)	59 (18.00)	286 (20.0)
Level of education		*p* < 0.0001	
General/primary studies	166 (9.5)	49 (15.0)	117 (8.2)
Secondary education (baccalaureate degree)	375 (21.4)	80 (24.5)	295 (20.7)
Post-secondary studies	146 (8.3)	15 (4.6)	131 (9.2)
Higher education (bachelor’s degree)	596 (34.0)	115 (35.2)	481 (33.7)
Postgraduate studies (master’s degree, residency, doctorate, other specializations)	471 (26.9)	68 (20.8)	403 (28.2)
Employment status		*p* < 0.0001	
Unemployed	23 (1.3)	7 (2.1)	16 (1.1)
Socially assisted	2 (0.1)	0 (0.0)	2 (0.1)
Householder	91 (5.2)	5 (1.5)	86 (6.0)
Retired	90 (5.1)	21 (6.4)	69 (4.8)
Student	577 (32.9)	138 (42.2)	439 (30.8)
Teleworking	47 (2.7)	9 (2.8)	38 (2.7)
I go to work every day	767 (43.7)	106 (32.4)	661 (46.3)
I work in a mixed regime (telework and commuting)	157 (9.0)	41 (12.5)	116 (8.1)
Body mass index (BMI)		*p* < 0.0001	
Underweight category (<18.5)	122 (7.0)	5 (1.5)	117 (8.2)
Normal weight (18.5–24.9)	955 (54.4)	143 (43.7)	812 (56.9)
Overweight category (25–29.9)	435 (24.8)	118 (36.1)	317 (22.2)
Obese (≥30)	242 (13.8)	61 (18.7)	181 (12.7)

**Table 2 nutrients-15-03841-t002:** Caloric intake of some sweet non-alcoholic drinks.

The Type of Drink	Calories per Serving	Total Carbohydrate Content	Sweeteners	Reference
Fresh orange	41 kcal (100 mL)	9.1 g	Natural carbohydrates	[49]
Fresh red orange	34 kcal (100 mL)	9.4 g	Natural carbohydrates	[50]
Coca-Cola	37 kcal (100 mL)	9.56 g	High fructose corn syrup, Sugars, Natural Flavors	[51,52]
Coca-Cola Caffeine Free	140 kcal (354 mL/12 fl oz)	39 g	High fructose corn syrup, Sugars, Natural Flavors	[53]
Coca-Cola Zero Sugar	0 kcal (221 mL/7.5 fl oz)	0 g	Aspartam, Acesulfame, Natural Flavors	[54]
Pepsi Max	0.4 kcal (100 mL)	0 g	Aspartame, Acesulfame K	[55]
Diet Pepsi	0.6 kcal (100 mL)	0 g	Aspartame, Acesulfame K	[56]
Pepsi	18 kcal (100 mL)	4.6 g	Acesulfame K, Sucralose, Sugars	[57]
Sprite	40 kcal (100 mL)	10.1 g	Sugars	[58]
Fanta Orange	100 kcal (221 mL/7.5 fl oz)	28 g	High fructose corn syrup, Sugars	[59]
Red Bull	43 kcal (100 mL)	10.2 g	Sugars, Natural and Artificial Flavors	[60]
Monster	47 kcal (100 mL)	11.3 g	Acesulfame K, Sucralose, Sugars	[61]
Gatorade G	26 kcal (100 mL)	6.43 g	Sugars, Natural and Artificial Flavors	[62]
Prigat Grapefruit	38 kcal (100 mL)	9.17g	Aspartame, Sugars	[63]
Lipton Iced Tea Peach	19 kcal (100 mL)	5.01 g	High fructose corn syrup, Acesulfame K Sucralose	[64]

**Table 3 nutrients-15-03841-t003:** Frequency of soft drink and alcoholic beverage consumption by age and gender.

Frequency of Soft-Drinks Consumption	Age								Gender			
	18–25		26–35		36–50		>50		Female		Male	
	*n*	(%)	*n*	(%)	*n*	(%)	*n*	(%)	*n*	(%)	*n*	(%)
	(A)		(B)		(C)		(D)		(A)		(B)	
Coffee	χ^2^ = 278.84, *p* < 0.0001								χ^2^ = 34.12, *p* < 0.0001			
Daily	269	43.9%	220 ^A^	64.5%	430 ^A,B^	83.8%	245 ^A,B^	85.4%	992 ^B^	69.5%	172	52.6%
2–3 times a week	132 ^B,C,D^	21.5%	26 ^D^	7.6%	21	4.1%	8	2.8%	138	9.7%	49 ^A^	15.0%
Once a week	32 ^C^	5.2%	12	3.5%	4	0.8%	5	1.7%	39	2.7%	14	4.3%
2–3 times a month	34	11.8%	137	22.3%	63	18.5%	78	15.2%	29	2.0%	10	3.1%
Very rarely or not at all	27 ^C,D^	9.4%	158 ^C,D^	25.8%	74	21.7%	52	10.1%	229	16.0%	82 ^A^	25.1%
Coca-Cola or Pepsi (or other variants)	χ^2^ = 195.03, *p* < 0.0001								χ^2^ = 40.44, *p* < 0.0001			
Daily	56 ^D^	9.1%	36 ^D^	10.6%	47 ^D^	9.2%	5	1.7%	110	7.7%	34	10.4%
2–3 times a week	120 ^C,D^	19.6%	50	14.7%	46	9.0%	16	5.6%	159	11.1%	73 ^A^	22.3%
Once a week	102 ^D^	16.6%	48 ^D^	14.1%	61 ^D^	11.9%	8	2.8%	174	12.2%	45	13.8%
2–3 times a month	137 ^C,D^	22.3%	63 ^D^	18.5%	78	15.2%	34	11.8%	255	17.9%	57	17.4%
Very rarely or not at all	198	32.3%	144^A^	42.2%	281^D^	54.8%	224 ^A,B,C^	78.0%	729 ^B^	51.1%	118	36.1%
Energy drinks	χ^2^ = 155.4, *p* < 0.0001								χ^2^ = 28.23, *p* < 0.0001			
Daily	19 ^D^	3.1%	7	2.1%	4	0.8%	1	0.3%	21	1.5%	10	3.1%
2–3 times a week	58 ^B,C,D^	9.5%	6	1.8%	9	1.8%	1	0.3%	50	3.5%	24 ^A^	7.3%
Once a week	35 ^C,D^	5.7%	8	2.3%	12	2.3%	2	0.7%	41	2.9%	16	4.9%
2–3 times a month	54 ^C,D^	8.8%	15 ^D^	4.4%	10	1.9%	2	0.7%	56	3.9%	25 ^A^	7.6%
Very rarely or not at all	447	72.9%	305 ^A^	89.4%	478 ^A^	93.2%	281 ^A,B,C^	97.9%	1259 ^B^	88.2%	252	77.1%
Green tea	χ^2^ = 12.65, *p* = 0.394								χ^2^ = 8.84, *p* = 0.065			
Daily	15	2.4%	12	3.5%	12	2.3%	10	3.5%	38	2.7%	11	3.4%
2–3 times a week	37	6.0%	13	3.8%	28	5.5%	20	7.0%	81	5.7%	17	5.2%
Once a week	29	4.7%	7	2.1%	15	2.9%	7	2.4%	45	3.2%	13	4.0%
2–3 times a month	82	13.4%	46	13.5%	63	12.3%	41	14.3%	174	12.2%	58 ^A^	17.7%
Very rarely or not at all	450	73.4%	263	77.1%	395	91.3%	209	72.8%	1089 ^B^	76.3%	228	69.7%
Black tea	χ^2^ = 29.17, *p* = 0.004								χ^2^ = 10.43, *p* = 0.034			
Daily	15	2.4%	9	2.6%	7	1.4%	2	0.7%	21	1.5%	12 ^A^	3.7%
2–3 times a week	28 ^C^	4.6%	9	2.6%	8	1.6%	6	2.1%	38	2.7%	13	4.0%
Once a week	17	2.8%	8	2.3%	15	2.9%	3	1.0%	33	2.3%	10	3.1%
2–3 times a month	51	8.3%	34	10.0%	29	5.7%	14	4.9%	101	7.1%	27	8.3%
Very rarely or not at all	502	81.9%	281	82.4%	454 ^A^	88.5%	262 ^A,B^	91.3%	1234 ^B^	86.5%	265	81.0%
Carbonated or sweetened drinks (1 serving = 330 mL, a glass)	χ^2^ = 225,73, *p* < 0.0001								χ^2^ = 27.74, *p* < 0.0001			
Daily, more than one serving	71 ^D^	11.6%	40 ^D^	11.7%	38 ^D^	7.4%	6	2.1%	116	8.1%	39 ^A^	11.9%
Daily, one serving	32	5.2%	18	5.3%	32 ^D^	6.2%	6	2.1%	64	4.5%	24 ^A^	7.3%
More than 2 times a week	103 ^C,D^	16.8%	38 ^D^	11.1%	46	9.0%	15	5.2%	154	10.8%	48 ^A^	14.7%
Twice a week	102 ^B,C,D^	16.6%	31	9.1%	33	6.4%	14^,^	4.9%	135	9.5%	45 ^A^	13.8%
Once a week	105 ^D^	17.1%	74 ^C,D^	21.7%	73^D^	14.2%	17	5.9%	221	15.5%	48	14.7%
Very rarely or not at all	200	32.6%	140	41.1%	291 ^A,B^	56.7%	229 ^A,B,C^	79.8%	737 ^B^	51.6%	123	37.6%
Alcoholic beverages (1 glass wine = 125 mL, 1 glass pure alcohol = 50 mL)		χ^2^ = 45.37, *p* < 0.0001							χ^2^ = 84.25, *p* < 0.0001			
Daily, more than one serving	7	1.1%	6	1.8%	4	0.8%	7	2.4%	5	0.4%	19 ^A^	5.8%
Daily, one serving	3	0.5%	4	1.2%	16 ^A^	3.1%	7	2.4%	21	1.5%	9	2.8%
More than 2 times a week	31	5.1%	10	2.9%	39 ^B^	7.6%	18	6.3%	67	4.7%	31 ^A^	9.5%
Twice a week	26	4.2%	29 ^A^	8.5%	43 ^A^	8.4%	19	6.6%	241	16.9%	53	16.2%
Once a week	89	14.5%	68	19.9%	87 ^A^	17.0%	50	17.4%	86	6.0%	31	9.5%
Very rarely or not at all	457 ^B,C,D^	74.6%	224	65.7%	324	63.2%	186	64.8%	1007 ^A^	70.6%	184	56.3%

Values with different superscript letters in a column (A,B,C,D) are significantly different (*p* < 0.05).

**Table 4 nutrients-15-03841-t004:** Results of the multinomial logistic regression for quantity of coffee consumption.

Independent Variables	1 Cup per Day			2–3 Cups per Day			More than 3 Cups per Day		
	OR	95% CI	*p*	OR	95% CI	*p*	OR	95% CI	*p*
Gender									
Male	1			1			1		
Female	1.776	(1.266–2.491)	0.001	1.547	(1.075–2.227)	0.019	1.000	(0.518–1.938)	0.999
Age (years)									
18–25	0.220	(0.138–0.315)	<0.0001	0.150	(0.091–0.248)	<0.0001	0.192	(0.074–0.493)	0.001
26–35	0.317	(0.193–0.521)	<0.0001	0.298	(0.176–0.503)	<0.0001	0.483	(0.187–1.249)	0.133
36–50	0.717	(0.438–1.173)	0. 185	1.072	(0.647–1.777)	0.786	1.438	(0.592–3.494)	0.423
>50	1			1			1		
Body mass index (BMI)									
Underweight	1			1			1		
Normal weight	1.269	(0.812–1.986)	0.296	1.644	(0.990–2.730)	0.055	1.265	(0.424–3.777)	0.673
Overweight	1.959	(1.192–3.220)	0.008	3.016	(1.739–5.231))	<0.0001	1.961	(0.572–6.716)	0.284
Obese	1.457	(0.849–2.500)	0.173	2.299	(1.273–4.150)	0.006	1.467	(0.439–4.904)	0.534
Smoking									
Occasionally	2.226	(1.372–3.612)	0.001	3.056	(1.829–5.106)	<0.0001	5.571	(2.311–8.426)	<0.0001
1–2 cigarettes daily	1.854	(1.029–3.342)	0.040	2.508	(1.342–4.688)	0.004	0.908	(0.115–4.167)	0.927
Excessive daily	3.972	(2.474–6.379)	<0.0001	9.165	(5.696–14.748)	<0.0001	9.697	(5.179–15.2846)	<0.0001
No	1			1			1		
Slipping time (hours)									
I have frequent insomnia	1.194	(0.634–2.250)	0.582	1.851	(0.991–3.456)	0.053	6.075	(2.405–15.347)	<0.0001
Under 7 h per night	0.958	(0.732–1.253)	0.754	1.248	(0.946–1.645)	0.117	3.006	(1.727–5.231)	<0.0001
Over 9 h per night	0.358	(0.188–0.684)	0.020	0.370	(0.182–0.752)	0.006	0.405	(0.053–3.122)	0.386
7–9 h per night	1			1			1		

Dependent variable: Rarely or not at all as reference category. Reference category: Gender: Male, Age: >50, BMI: Underweight, Smoking: No, Sleep: 7–9 h per night.

**Table 5 nutrients-15-03841-t005:** Results of the multinomial logistic regression for quantity of Coca-Cola or Pepsi consumption.

Independent Variables	Up to 500 mL per Day			Up to 1 L per Day			Over 1 L per Day		
	OR	95% CI	*p*	OR	95% CI	*p*	OR	95% CI	*p*
Gender									
Male	1			1			1		
Female	0.668	(0.514–0.867)	0.0021	0.391	(0.234–0.653)	<0.0001	0.495	(0.243–1.009)	0.053
Age (years)									
18–25	2.1	(1.448–3.045)	<0.0001	5.369	(3.613–7.978)	<0.0001	6.524	(2.208–9.677)	0.006
26–35	1.534	(1.012–2.324)	0.044	4.347	(2.841–6.651)	<0.0001	5.289	(1.971–8.589)	0.009
36–50	1.31	(0.893–1.923)	0.168	3.279	(2.184 –4.922)	<0.0001	4.429	(0.530–7.023)	0.17
>50	1			1			1		
Body mass index (BMI)									
Underweight	1			1			1		
Normal weight	1.021	(0.667–1.562)	0.925	1.622	(0.539–3.884)	0.39	1.817	(0.395–8.351)	0.443
Overweight	1.425	(0.888–2.287)	0.142	2.934	(0.913–9.439)	0.071	2.648	(0.508–13.798)	0.248
Obese	1.903	(1.122–3.229)	0.017	5.064	(1.473–14.411)	0.01	5.29	(1.572–15.721)	0.013
Smoking									
Occasionally	1.702	(1.190–2.435)	0.004	1.818	(0.785–4.210)	0.163	1.818	(0.608–5.434)	0.285
1–2 cigarettes daily	2.213	(1.430–3.423)	<0.0001	2.263	(0.887–6.293)	0.085	1.031	(0.586–1.778)	0.942
Excessive daily	2.746	(2.096–3.598)	<0.0001	4.59	(2.693–7.823)	<0.0001	4.76	(2.408–9.409)	<0.0001
No smoke	1			1			1		
Slipping time (hours)									
I have frequent insomnia	1.161	(0.705–1913)	0.558	2.932	(1.296–6.633)	0.01	5.56	(2.143–14.683)	<0.0001
Under 7 h per night	1.332	(1067–1.6621)	0.011	1.478	(0.890–2.457)	0.131	3.908	(1.737–8.789)	0.001
Over 9 h a night	1.234	(0.674–2.257)	0.496	0.573	(0.076–4.307)	0.588	1.416	(0.853–2.321)	0.186
7–9 h per night	1			1			1		

Dependent variable: Rarely or not at all as reference category. Reference category: Gender: Male, Age: >50, BMI: Underweight, Smoking: No, Sleep: 7–9 h per night.

**Table 6 nutrients-15-03841-t006:** Results of the multinomial logistic regression for quantity of energy drinks consumption.

Independent Variables	1 Dose per Day			2 Doses per Day			Over 2 Doses per Day		
	OR	95% CI	*p*	OR	95% CI	*p*	OR	95% CI	*p*
Gender									
Male	1		1			1			
Female	0.515	(0.333–0.798)	0.003	0.761	(0.442–1.310)	0.324	0.289	(0.131–0.637)	0.002
Age (years)									
18–25	4.149	(2.981–12.191)	<0.0001	1.764	(0.876–3.551)	0.112	2.421	(0.640–9.162)	0.193
26–35	3.276	(1.057–10.149)	0.04	1.364	(0.647–2.878)	0.414	1.081	(0.244–4.799)	0.918
36–50	3.473	(1.167–10.333)	0.025	1.136	(0.566–2.281)	0.72	1.186	(0.294–4.796)	0.81
>50	1			1			1		
Smoking									
Occasionally	2.162	(1.225–3.815)	0.008	2.313	(1.228–4.355)	0.009	1.247	(0.354–4.388)	0.731
1–2 cigarettes daily	2.815	(1.820–4.353)	0.001	0.824	(0.250–2.720)	0.751	0.827	(0.107–6.398)	0.855
Excessive daily	3.168	(1.643–6.112)	<0.0001	2.087	(1.251–3.481)	0.005	1.663	(0.713–3.880)	0.239
No smoke	1			1			1		

Dependent variable: rarely or not at all as reference category. Reference category: Gender: Male, Age: >50, Smoking: No.

## Data Availability

Not applicable.

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
