# Peer review of "Evaluation of Non-Alcoholic Beverages and the Risk Related to Consumer Health among the Romanian Population"

_nutrients, 2023, doi:10.3390/nu15173841_

Round 1
Reviewer 1 Report
This paper analyzed the effect of several kinds of non-alcoholic beverages on the human risk in Romania. The data were abundant and the content was adequate. The results were also meaningful. I suggested that this paper can be accepted after minor revision.
1. The “Introduction” should be shortened due to its tedious organization.
2. To make the investigation procedure of this study more clear, more detailed information should be added in the “2.1. Study Design”.
3. Table 1 should be separated to several tables based on the data characteristics.
4. Figure 5 is not clear, please replace by a new one with higher resolution.
5. Figures 7 and 8 are not clear, please replace by new ones with higher resolution.
6. Figure 15 is not clear, please replace by a new one with higher resolution.
7. Figure 18 is not clear, please replace by a new one with higher resolution.
8. More data should be presented in the conclusion.
9. Please check and unify the format of references in accordance the journal guideline.
Author Response
Thank you for your kindness and effort in evaluating the manuscript, especially for the useful advice in improving the manuscript.
- The “Introduction” should be shortened due to its tedious organization.
We reduced the text allocated to the introduction
- To make the investigation procedure of this study more clear, more detailed information should be added in the “2.1. Study Design”.
We added:
The questionnaire was structured in three main parts: the first part aimed at obtaining socio-demographic and anthropometric information (age, gender, residence, level of education, occupational status, weight and height), the second part aimed at collecting data related to the frequency of consumption of different categories of non-alcoholic beverages, the amount consumed, the types of associated sweeteners, and in the part of the third was the collection of information related to lifestyle (physical activity, tobacco consumption, alcohol consumption, duration and quality of sleep, periodicity of health assessment, type of professional activity) but also a series of aspects that alter the quality of life ( the state of the immune system, the presence of conditions such as fatigue, nervousness, depression, anxiety, migraines, agitation, palpitations, compulsive eating or loss of appetite), as well as the identification of consumption addictions generated by some non-alcoholic beverages.
- Table 1 should be separated to several tables based on the data characteristics.
We changed
- Figure 5 is not clear, please replace by a new one with higher resolution.
We changed
- Figures 7 and 8 are not clear, please replace by new ones with higher resolution.
We changed
- Figure 15 is not clear, please replace by a new one with higher resolution.
We changed
- Figure 18 is not clear, please replace by a new one with higher resolution.
We changed
- More data should be presented in the conclusion.
We added
Also, the addiction generated by a series of non-alcoholic drinks (coffee, sweetened carbonated and non-carbonated drinks) must also be managed, excessive consumption often taking place at the expense of proper hydration with water, which can lead to an aggravation of imbalances in the body consumers (dehydration, improper detoxification, blood thickening, etc.). Managing the consumption of sweetened beverages must be done from a young age to prevent the development of excessive consumption habits. The results of the study indicate a significant addiction to coffee consumption, but also an increased frequency of states of fatigue and agitation generated by inadequate rest and probably by excess caffeine. In order to counterbalance the negative effects of excessive consumption of carbohydrates and caffeine, proper hydration of the body and stimulation of physical activity are required.
- Please check and unify the format of references in accordance the journal guideline.
We corrected
Reviewer 2 Report
General comment
In my opinion, this study is an exhaustive and excellent work of research that deals with one of the most serious and silent health problems of this century, obesity and the excessive consumption of stimulating substances. However, after reading it carefully, I would like to comment some aspects before being considered for publication.
Text
1. Abstract. The abstract should contain a brief final reflection about the meaning of the study and its relevance. That is, how the discoveries made can help solve the problem of excessive consumption of unhealthy non-alcoholic drinks.
2. Lines 130-132. In the sentence “In general, a typical dose…the content of a cup of tea” the word “energy” should be replaced by the word “stimulant” or something similar, since from my point of view the term “energy” is more correct when we refer to the contribution and/or consumption of calories.
3. Lines 391-395. The idea that respondents who consumed a large amount of coffee per day suffered effects related to problems with the quality of sleep seems to be repeated.
4. Lines 562-564. The text between these lines is not well understood; perhaps it would be appropriate to rewrite it.
5. Lines 641-642. The word “and” is missing between “attention” and “power of concentration”.
6. Lines 643-645. The text between these lines should be rewritten; it is understandable but it sounds strange.
7. Conclusions section. This section exposes the consequences of consuming unhealthy non-alcoholic beverages due to their high levels of sugars, sweeteners, and/or caffeine. However, from my point of view, emphasis should be placed on this aspect based on the results obtained with the Romanian population. That is, to show which aspects are the most relevant regarding the habits found in the study, their consequences, and the possible measures to solve it.
Table
1. Table 1. The body mass index (BMI) is a value that can yield unreliable data depending on the muscular development of the individual, since the weight of the muscle contributes to raising this index, but it does not mean that the person is overweight or obese. This aspect may be even more relevant today due to the cult that exist for developing a good muscular and physical tone, especially among young people. A simple type of test to better measure the level of body fat is the measurement of skin folds. However, this and other type of measurements of body fat are more expensive to carry out and therefore less used. Even so, it would be advisable to make a mention of this in the manuscript to make clear to the reader the possible lack of reliability that may exist in the data.
2. Table 3. I think it would be better to leave more space between rows because otherwise reading the data becomes a little complicated.
Figures
1. Figures 2, 5, 7, 8, 15, 18, and 21. The image quality of these figures, especially those of 15, 18, and 21, should be improved. It is difficult to visualize the displayed data.
Author Response
Thank you for your kindness and effort in evaluating the manuscript, especially for the useful advice in improving the manuscript.
- The abstract should contain a brief final reflection about the meaning of the study and its relevance. That is, how the discoveries made can help solve the problem of excessive consumption of unhealthy non-alcoholic drinks.
We added
Reducing the excessive consumption of sweetened drinks can be achieved through an appropriate consumption of water, fruits and by intensifying physical activity as a way of counterbalancing the excess caloric intake.
- Lines 130-132. In the sentence “In general, a typical dose…the content of a cup of tea” the word “energy” should be replaced by the word “stimulant” or something similar, since from my point of view the term “energy” is more correct when we refer to the contribution and/or consumption of calories.
We deleted the paragraph as a suggestion to shorten the introduction.
3.Lines 391-395. The idea that respondents who consumed a large amount of coffee per day suffered effects related to problems with the quality of sleep seems to be repeated.
We changed
Excessive coffee consumption can affect the quality of sleep, respondents who have frequent insomnia are more likely to drink more than 3 cups of coffee per a day (OR=6.075) compared to people who sleep 7-9 hours a night. The most pronounced tendency to consume 2-3 cups of coffee a day or even more is the people who sleep less hours a night or have repeated episodes of insomnia.
- Lines 562-564. The text between these lines is not well understood; perhaps it would be appropriate to rewrite it.
We changed
According to the socio-demographic data, the majority of respondents live in the urban environment (80.3% of them) and have higher education (60.9%), are women (81.4%) and commute daily to work (43 .7%).
- Lines 641-642. The word “and” is missing between “attention” and “power of concentration”.
We added
- Lines 643-645. The text between these lines should be rewritten; it is understandable but it sounds strange.
We changed
The state of well-being and implicitly the quality of life is affected although most of the respondents are up to 35 years old, they declared that they face a series of problems such as: fatigue, nervousness, agitation, depression, very few of the participants in the study declared that they feel well (Figure 19).
- Conclusions section. This section exposes the consequences of consuming unhealthy non-alcoholic beverages due to their high levels of sugars, sweeteners, and/or caffeine. However, from my point of view, emphasis should be placed on this aspect based on the results obtained with the Romanian population. That is, to show which aspects are the most relevant regarding the habits found in the study, their consequences, and the possible measures to solve it.
Also, the addiction generated by a series of non-alcoholic drinks (coffee, sweetened carbonated and non-carbonated drinks) must also be managed, excessive consumption often taking place at the expense of proper hydration with water, which can lead to an aggravation of imbalances in the body consumers (dehydration, improper detoxification, blood thickening, etc.). Managing the consumption of sweetened beverages must be done from a young age to prevent the development of excessive consumption habits. The results of the study indicate a significant addiction to coffee consumption, but also an increased frequency of states of fatigue and agitation generated by inadequate rest and probably by excess caffeine. In order to counterbalance the negative effects of excessive consumption of carbohydrates and caffeine, proper hydration of the body and stimulation of physical activity are required.
Table
- Table 1. The body mass index (BMI) is a value that can yield unreliable data depending on the muscular development of the individual, since the weight of the muscle contributes to raising this index, but it does not mean that the person is overweight or obese. This aspect may be even more relevant today due to the cult that exist for developing a good muscular and physical tone, especially among young people. A simple type of test to better measure the level of body fat is the measurement of skin folds. However, this and other type of measurements of body fat are more expensive to carry out and therefore less used. Even so, it would be advisable to make a mention of this in the manuscript to make clear to the reader the possible lack of reliability that may exist in the data.
We added
For a good accuracy of the clinical results, BMI must be correlated with the percentage of fat, because there are situations in performance athletes when a higher BMI does not represent an excess of adipose tissue but a higher percentage of muscle mass. In the case of a predominantly sedentary behavior, excess weight is accompanied by excess adipose tissue.
- Table 3. I think it would be better to leave more space between rows because otherwise reading the data becomes a little complicated.
We separated with lines
Figures
- Figures 2, 5, 7, 8, 15, 18, and 21. The image quality of these figures, especially those of 15, 18, and 21, should be improved. It is difficult to visualize the displayed data.
We changed